# Human Endogenous Retrovirus Expression Is Associated with Head and Neck Cancer and Differential Survival

**DOI:** 10.3390/v12090956

**Published:** 2020-08-28

**Authors:** Allison R. Kolbe, Matthew L. Bendall, Alexander T. Pearson, Doru Paul, Douglas F. Nixon, Marcos Pérez-Losada, Keith A. Crandall

**Affiliations:** 1Computational Biology Institute, Milken Institute School of Public Health, George Washington University, Washington, DC 20052, USA; akolbe@email.gwu.edu (A.R.K.); mlosada@gwu.edu (M.P.-L.); 2Division of Infectious Diseases, Department of Medicine, Weill Cornell Medicine, New York, NY 10021, USA; mlb4001@med.cornell.edu (M.L.B.); dnixon@med.cornell.edu (D.F.N.); 3Department of Medicine, The University of Chicago Medicine, Chicago, IL 60637, USA; apearson5@medicine.bsd.uchicago.edu; 4Division of Hematology and Medical Oncology, Weill Cornell Medicine, New York, NY 10021, USA; dop9054@med.cornell.edu; 5Department of Biostatistics and Bioinformatics, Milken Institute School of Public Health, George Washington University, Washington, DC 20052, USA; 6CIBIO-InBIO, Centro de Investigação em Biodiversidade e Recursos Genéticos, Universidade do Porto, Campus Agrário de Vairão, 4485-661 Vairão, Portugal

**Keywords:** RNA-seq, transposable element, endogenous retrovirus, cancer, TCGA

## Abstract

Human endogenous retroviruses (HERVs) have been implicated in a variety of human diseases including cancers. However, technical challenges in analyzing HERV sequence data have limited locus-specific characterization of HERV expression. Here, we use the software Telescope (developed to identify expressed transposable elements from metatranscriptomic data) on 43 paired tumor and adjacent normal tissue samples from The Cancer Genome Atlas Program to produce the first locus-specific retrotranscriptome of head and neck cancer. Telescope identified over 3000 expressed HERVs in tumor and adjacent normal tissue, and 1078 HERVs were differentially expressed between the two tissue types. The majority of differentially expressed HERVs were expressed at a higher level in tumor tissue. Differentially expressed HERVs were enriched in members of the HERVH family. Hierarchical clustering based on HERV expression in tumor-adjacent normal tissue resulted in two distinct clusters with significantly different survival probability. Together, these results highlight the importance of future work on the role of HERVs across a range of cancers.

## 1. Introduction

Human endogenous retroviruses (HERVs) make up approximately 8% of the human genome, but their role in disease remains poorly understood [1,2,3,4,5]. Expression of human endogenous retroviruses is altered in numerous cancers, including melanoma [6,7], breast cancer [8,9,10], and ovarian cancer [11]. Members of multiple HERV families, including HERVH, HERVK, HERVF, HERVR, and HERVS, have been identified in cancer cell lines [12]. HERVs have been linked to oncogenesis at the DNA and protein level, but there is also evidence of beneficial HERV effects [13]. Therefore, the relationship between HERV expression and cancer is complex, and may have both positive and negative effects on cancer progression and clinical outcomes. Although many studies have characterized HERV expression in cancer and other diseases, the high degree of repetitive and highly similar sequences in HERV elements have made locus-specific characterization of HERVs a significant challenge. Thus, characterizing the cancer retrotranscriptome has remained an elusive but important goal for cancer research.

Head and neck squamous cell carcinoma (HNSCC) affects more than 700,000 people per year worldwide, with a mortality rate that exceeds 50% [14]. HNSCC typically originates from the epithelial tissues of the oral cavity, larynx, oropharynx, or hypopharynx. HNSCC is a highly heterogeneous disease, with distinct subtypes of different etiologies and presenting with different molecular changes [15]. In particular, human papillomavirus (HPV) status, which is a risk factor for developing HNSCC, has also been shown to result in a distinct subtype of HNSCC [16]. Several existing studies have examined the transcriptome in HNSCC [17,18,19] and reviewed in Leemans et al. [15] using protein-coding and non-coding genomic annotations. A few studies have examined HERV family-level expression in the pan-cancer The Cancer Genome Atlas (TCGA, https://www.cancer.gov/about-nci/organization/ccg/research/structural-genomics/tcga) cohort [20,21,22,23]. Yet no study to date has examined the implications of locus-specific HERV expression specific to HNSCC.

Here, we apply Telescope [24], a computational software pipeline which provides accurate estimation of transposable element expression resolved to specific genomic locations, to 43 paired tumor and tumor-adjacent normal tissue RNA-seq datasets available from TCGA [25]. This work provides the first locus-specific retrotranscriptomic analysis of HNSCC.

## 2. Materials and Methods

### 2.1. TCGA Data

Paired-end RNA-sequencing data were obtained from TCGA [25]. A total of 43 head and neck cancer cases with paired tumor and healthy tissue samples were downloaded from the Genomic Data Commons (GDC) Data Portal [26]. Corresponding clinical data, including tumor stage, tumor location, vital status, and tobacco history, were downloaded using the R package *TCGAbiolinks* [27] and further supplemented by clinical data provided in [17].

### 2.2. Patient Demographics

The majority of the cancer cases were from the oral cavity (32/43); the remainder of the tumors were laryngeal in origin (11/43). Although HNSCC also occurs in the oropharynx, none of the available paired RNA-seq samples originated from oropharyngeal tumors. Tumor samples represented stages I–IVa (stage I: 2; stage II: 15; stage III: 8; stage IVa: 17; not reported: 1). Patients were majority male (29/43), and mostly current or former smokers. Patients without tobacco history data (8/43) were categorized as smokers, as in [17].

### 2.3. Sequence Data Quality Control and Trimming

Files were downloaded from GDC as BAM files (*.bam) and were converted to FASTQ (*.fastq) for quality control and trimming using the SamToFastq tool in Picard version 2.6 [28]. The downloaded files contained between 37,379,141 and 116,430,322 reads, with an average of 73,776,707 ± 1,841,593 (mean ± standard error). Illumina adapters were removed and reads were trimmed for quality using Trimmomatic version 0.33 [29], removing leading and trailing bases below quality 3 and a 4-base sliding window with a quality threshold of 15. Reads with a length less than 36 bases after trimming were discarded. On average, 95% of reads remained after trimming, for a final range of 34,997,325 to 109,343,593 trimmed reads per sample.

### 2.4. Retrotranscriptome Quantification

Trimmed reads were aligned to the human genome (hg38) using Bowtie2 [30], as described in [24]. Briefly, the Bowtie2 alignment options were set to perform a local alignment search (*--very-sensitive-local*), allowing up to 100 alignments per read (*-k 100*), with a minimum alignment score threshold such that fragments with ~95%+ sequence identity would be reported. Overall alignment rates ranged from 86 to 93%. Bowtie2 alignments were then provided to the Telescope v1.0.2 assign module using the HERV annotation provided by [24] and theta prior of 200,000. Final count numbers were loaded into DESeq2 [31] using the *DESeqDataSetFromMatrix* function, with tissue origin (larynx or oral cavity) and tissue type (normal or tumor) as variables in the model formula. The variance-stabilizing transformation was used for principal component and clustering analysis. Normalization and differential expression analysis were performed using the *DESeq* function in DESeq2 [31], which implements a negative binomial model and Wald test. A false-discovery rate (FDR) threshold of 0.05 was used for HERV expression. The full output of DESeq2 analysis of differential HERV expression is provided in Appendix A.

Hierarchical clustering of the top 100 differentially expressed HERVs was performed using pvclust [32]. Chi-squared tests were performed between the two tumor clusters and between the two normal clusters to determine statistical differences in gender, tissue type, smoking history, or early (stage I–II) vs. late (stage III–IVa) tumor stage. Statistical significance was determined at *p* < 0.05.

### 2.5. Transcriptome Quantification

Gene expression was quantified using kallisto v0.43.1 [33] using the Ensembl v96 transcriptome assembly [34]. The kallisto index was built using default settings [33]. Quantification was performed using kallisto quant with default settings. The resulting abundance tables were imported into R using tximport [35] and the transcript-to-gene mapping file provided with the Ensembl v96 index files (https://github.com/pachterlab/kallisto-transcriptome-indices). The model formula was specified as described above, with tissue origin (larynx or oral cavity) and tissue type (normal or tumor). On average, kallisto pseudoaligned 95% of reads. The variance-stabilizing transformation was used for principal component and clustering analysis. Normalization and differential expression analysis were performed using the *DESeq* function in DESeq2 [31], which implements a negative binomial model and Wald test. A FDR threshold of 0.01 was used for gene expression. The full output of DESeq2 analysis of host differential expression is provided in Appendix A.

### 2.6. Determination of HPV Status

HPV status was evaluated using PathoScope 2.0 [36] because the TCGA metadata did not include HPV status on all patients from which RNA-seq data were collected. Trimmed reads were aligned to the representative and reference viroid and virus genome databases from GenBank (https://www.ncbi.nlm.nih.gov/genbank/) using the PathoScope 2.0 MAP module. Reads that aligned to the human genome (hg38) were filtered from the alignment. Taxonomic assignment was performed using the PathoScope 2.0 ID module. The R package taxonomizr [37] was used to retrieve full taxonomic lineages from taxonomy identifiers, and composition and taxonomic data were imported into the R package phyloseq [38] for further manipulation. HPV status was defined by the presence of >10 reads mapped to the genus *Alphapapillomavirus*. Two patients met the criteria for being HPV positive in this study.

### 2.7. Gene Set Enrichment Analysis

Gene set enrichment analysis (GSEA) [39] was performed to determine enrichment of HERV families in tumor tissue. A list of HERVs ranked by differential expression was calculated by sign(logFC)×−log_10_(adjusted *p*-value). A custom gene set for HERV families was created using the HERV annotation generated in [24]. The ranked gene list and custom gene set database were provided to GSEAPreranked (GSEA v4.0.0), and analysis was performed with 1000 permutations and the “classic” enrichment statistic. A FDR cutoff of 0.25 was used according to the author’s instructions [39].

### 2.8. Clustering and Survival Analysis

In order to identify and characterize subtypes of HNSCC, clustering was performed on variance-stabilized HERV expression data from tumor and normal tissue. Clustering was performed separately on each tissue type using a Euclidean distance matrix and the “ward.D2” method implemented in hclust [40]. In each case, clusters were identified using the function *cutree*. Tumor samples were divided into two clusters. One outlier sample was removed. Survival analysis was performed on the two clusters which incorporated 38 samples. Normal samples were divided into two clusters. Survival analysis was performed using the R package survival v2.44 [41,42]. According to the approach implemented in [17], survival times were censored at 5 years because most cancer-related events occur before that time. For patients recorded as alive, data were censored at last follow up. Patients without death or follow-up data were excluded from the analysis, which left 39 patients in total. Survival curves were fit using the Kaplan–Meier model, stratifying by cluster membership. Significance was determined with the log-rank test at a p-value threshold of 0.05, but because we conduct tests independently on normal and tumor clusters, we also employ the Benjamini–Hochberg [43] multiple test correction for the two clusters at the *p*-value threshold of 0.10. Such corrections have not been routinely performed in prior work on TCGA data [17,44,45]. Resulting curves were plotted using the *ggsurvplot* function in survminer [46].

## 3. Results

### 3.1. HERV Expression

Using Telescope [24], we identified 3520 HERVs expressed at a minimum of 5 counts in at least 2 samples across tumor and adjacent normal tissue from HNSCC cancer cases. Tumor tissue expressed significantly more HERVs than adjacent normal tissue; on average, 1846 HERVs were expressed in tumor samples compared to 1550 HERVs expressed in normal samples. The HERV expression pattern between tumor and normal tissue is distinct, as shown by principal component analysis (Figure 1A).

Out of 3520 identified HERVs, 1078 were differentially expressed between tumor and normal samples. Of these, the majority (802) were expressed at a higher level in tumor tissue compared to normal tissue (Figure 1B).

GSEA was performed to determine whether differentially expressed HERVs were enriched in particular HERV families. Among upregulated HERVs in tumor tissue, the HERVH and HARLEQUIN families were significantly enriched at *q* < 0.25 (Appendix A). The HERVE, HML6, PRIMA4, and HERVEA families were all significantly enriched (*q* < 0.25) among downregulated HERVs (Appendix A).

### 3.2. HERV Expression, Phenotypic and Genotypic Associations

The expression signature of top differentially expressed HERVs differed based on the tissue of origin. Clustering of the top 100 differentially expressed HERVs resulted in two distinct tumor clusters and two distinct normal clusters (Figure 2), which had significantly different tissue origins (chi-squared test, *p* < 0.05). In both cases, one cluster was dominated by oral cavity tumors (normal cluster: 10/10; tumor cluster: 22/23), which included mouth and tongue tumors, as well as tumors which overlapped the lip, oral cavity, and pharynx. The second cluster in both cases was split between oral cavity and laryngeal tumors (normal cluster: 19/30 oral cavity; tumor cluster: 13/23 oral cavity). The sublocation of oral cavity tumors varied between the two clusters; those which clustered with laryngeal tumors were more likely to be from the tongue; in contrast, the majority of oral cavity tumors in the other cluster corresponded to tumors which overlapped the lip, oral cavity, and pharynx. Smoking status, gender, age at diagnosis, and tumor stage (early vs. late) were not significantly different between the two tumor clusters or between the two normal clusters. Three normal tissue samples clustered with tumor samples. In all three cases, the paired tumor sample was found in the same cluster, and in two (HNSC06 and HNSC18), the tumor and normal sample clustered tightly together. These results may be indicative of tumor-like expression patterns in some tumor-adjacent tissues. Although HPV status is an important risk factor in HNSCC, only two patients from this cohort were HPV positive (samples HNSC16 and HNSC27). Interestingly, HPV-positive patients in this analysis did not cluster together, indicating that other variables were driving the expression patterns in these cases.

In order to compare HERV expression with non-HERV gene expression, we identified differentially expressed host genes with kallisto [33]. Among significantly differentially expressed genes were many genes identified in previous studies, including *TP63*, *CDKN2A*, and *FADD* [17,50,51]. Similar to HERV expression, principal component analysis showed distinct expression patterns between tumor and normal tissue (Appendix A). As previously described [17], many genes are differentially expressed between these tissue types. Out of 33,260 genes expressed in this dataset, 4348 were significantly upregulated and 4295 were significantly downregulated (FDR < 0.01) in tumor tissue (Appendix A). Differentially expressed genes were found throughout the genome, similar to patterns observed for HERVs (Figure 3).

### 3.3. Survival Analysis

In order to evaluate the impact of HERV expression patterns on subtypes of HNSCC which are associated with patient survival probability, we performed hierarchical clustering on the HERV expression data followed by survival analysis. For this analysis, hierarchical clustering was performed separately on tumor and normal tissue, using all HERV expression data.

Two clusters were identified from tumor tissue after removing a single outlier sample. Gender, smoking status, and tumor stage (early vs. late, as described previously) were not significantly different between these two clusters. Differences in tissue origin were significant, with one cluster composed entirely of oral cavity tumors, and the other split between laryngeal and oral cavity tumors (chi-squared test, *p* < 0.05). These clusters were also evident in the principal component analysis (Appendix A). For the two tumor clusters, probability of patient survival was not significantly related to cluster membership (Appendix A).

Similarly, two clusters were identified from normal tissue (Appendix A). However, gender, tumor stage, tissue origin, and smoking status were not significantly different between these clusters (chi-squared test, *p* < 0.05). Patient survival probability differed significantly between the two normal tissue clusters (Figure 4). There was no significant enrichment of any HERV family between the two clusters, as determined by GSEA.

## 4. Discussion

HERVs have been implicated in a wide range of cancers, including breast, prostate, lymphoma, melanoma, and ovarian cancers (reviewed in [13]). Much of the work linking HERVs and cancer has focused on the HERV-K family, the most active family of HERVs, many of which are full length or nearly full length [13]. HERVK (HML2) has even been identified as a biomarker for breast cancer [53,54]. Interestingly, HERVK (HML2) was not implicated in our analysis of HNSCC cancers, whereas HERVK (HML6) was identified as being significantly enriched in the downregulated set of HERVs, indicating that although the HERVK has been implicated in many other types of cancers, not all members of this superfamily of retroelements are involved in all cancers.

In our analysis, the HERVH family was significantly enriched in tumor tissue in HNSCC. Although HERVH is not the most commonly implicated HERV in cancer, HERVH has been identified in various cancer cell lines [12,55] and pan-cancer studies [23]. Kong et al. [23] observed overexpression of HERVH-internal, the proviral portion of HERVH, in six cancer types including HNSCC. HERVH is one of the most abundant HERV families, with thousands of HERVH elements in the human genome [56]. Despite its prevalence, very few HERVH genes encode intact envelope proteins [56]. Future work should assess whether HERVH is a bystander or serves a functional role in HNSCC.

Other HERV families were found at either higher levels (HARLEQUIN) or lower levels (HERVE, HML6, PRIMA4, and HERVEA) in tumor tissues compared to adjacent normal tissues. Little is known about the role of these HERV families in HNSCC or other cancers. Future work should explore potential roles of these smaller HERV families in HNSCC.

Here, we show that HERV expression patterns as defined by hierarchical clustering in tumor-adjacent normal tissue is related to altered probability of patient survival. A similar result was previously shown by [57], who identified two clusters of non-HERV gene expression in normal tissue which were associated with differential survival. Similar to our findings, differential survival was not observed when clustering expression from tumor tissue. Although it may seem counterintuitive that expression from tumor tissue does not predict patient survival and tumor-adjacent normal tissue does, similar results have been found by a number of research groups. Transcriptional profiles were analyzed from six TCGA datasets and found that tumor-adjacent normal tissue was more predictive of patient survival than tumor tissue [58]. The authors hypothesized that tumor-adjacent normal tissue may reflect a patient’s overall immunity or metabolic level, and therefore may be more informative of patient outcomes than genome dysregulation found in tumor tissue [58]. Tumor-adjacent normal tissue is known to be morphologically and phenotypically distinct from normal tissue and possesses molecular alterations that make it a unique intermediate between tumor tissue and true normal tissue [59]. Furthermore, gene expression in tumor-adjacent normal tissue has been used to predict survival in colorectal cancer patients [60], predict clinical outcome in breast cancer [61], and identify breast cancer subtypes [62]. Our findings provide additional evidence that not only are gene patterns distinct in tumor-adjacent normal tissue, but also HERV expression patterns. Therefore, future work should evaluate the links between HERV expression and other tumor characteristics which affect patient survival.

## 5. Conclusions

Here, we provide the first locus-specific analysis of HERV expression in head and neck cancer. We found that many HERV loci are expressed in both tumor and tumor-adjacent normal tissue, with many differentially expressed loci between the two tissue types. Interestingly, members of the HERVH family were expressed at higher levels in tumor tissue, whereas HERVK (HML2) expression, which has been associated with other types of cancer, did not vary. Furthermore, HERV expression patterns in tumor-adjacent normal tissue, as described by hierarchical clustering, were associated with differential survival. These findings emphasize the potential differences in HERV expression patterns between different cancers and emphasize the need for future work on the role of HERVs in head and neck cancer.

## Figures and Tables

**Figure 1 viruses-12-00956-f001:**
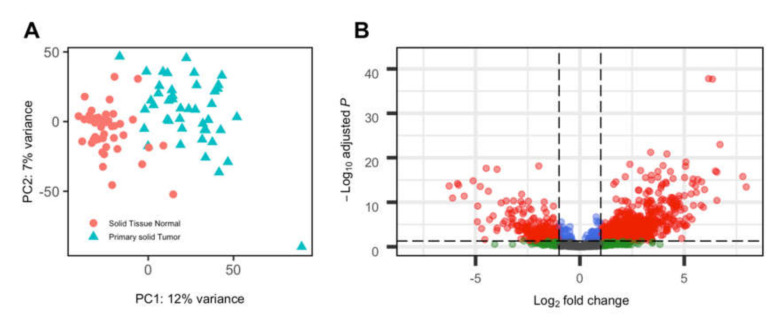
Human endogenous retrovirus (HERV) expression in head and neck squamous cell carcinoma (HNSCC) tumor and paired normal samples. (**A**) Principal component analysis, with shape and color indicating tumor vs normal tissue. (**B**) Volcano plot of differential expression between tumor and normal tissue. Positive log-fold change indicates higher expression in tumor tissue. The dashed horizontal line indicates adjusted *p*-value threshold of 0.05. The dashed vertical line indicates a log2-fold change of +/−1.5.

**Figure 2 viruses-12-00956-f002:**
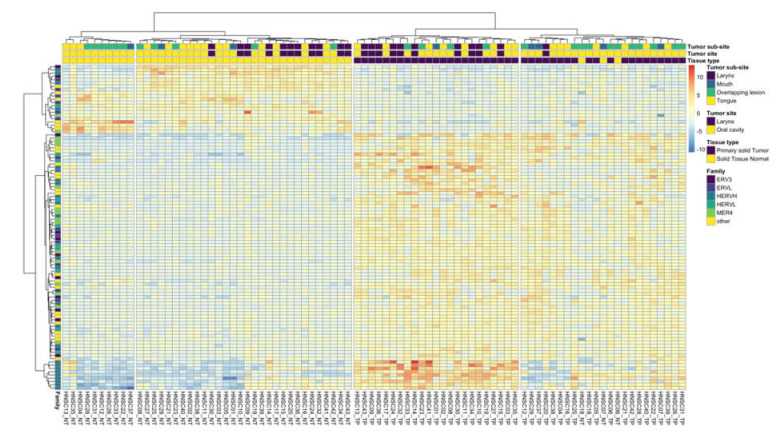
Expression for the top 100 differentially expressed HERVs. Heatmap colors represent the difference between each sample’s expression level and the mean expression level of each HERV (sample expression − mean expression). Row colors indicate HERV family membership. HERV families with five or fewer members present in the top 100 differentially expressed HERVs are presented as “other”. Clustering of the top 100 differentially expressed HERVs was performed on variance-stabilized expression data using the method ward.D2 in R [40]. An ordered list of the HERV loci shown in this figure is available in Appendix A. Heatmap was generated with pheatmap [47], with the viridis [48] and RColorBrewer color palette [49].

**Figure 3 viruses-12-00956-f003:**
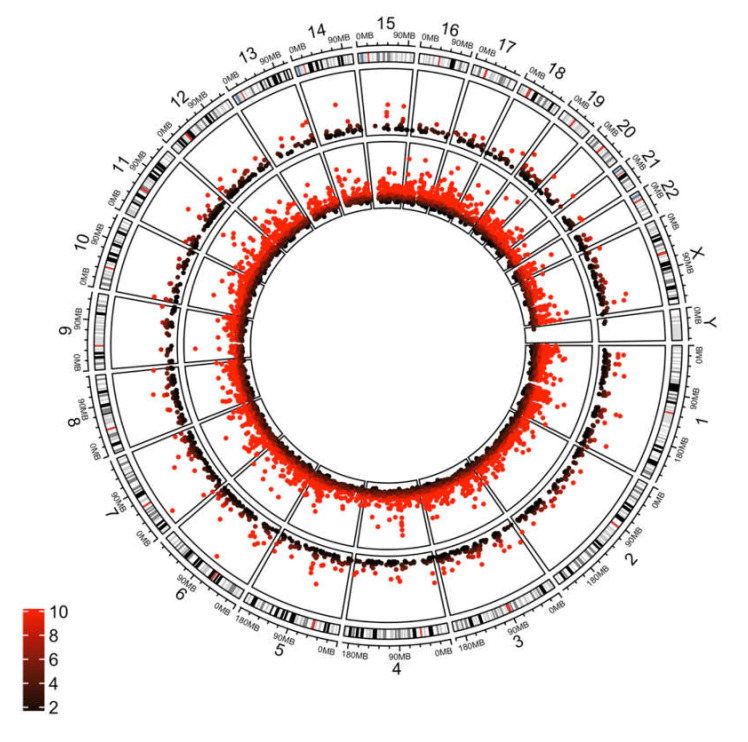
Circos plot showing differentially expressed host genes (inner ring) and HERVs (outer ring) across chromosomal locations. Values are plotted on a −log (adjusted *p*-value) scale. Color indicates degree of significance, with the darkest colors corresponding to lowest significant *p*-value (0.05), with the brightest red corresponding to an adjusted *p*-value of *p* < 0.00000000001. Plot was generated using circlize in R [52].

**Figure 4 viruses-12-00956-f004:**
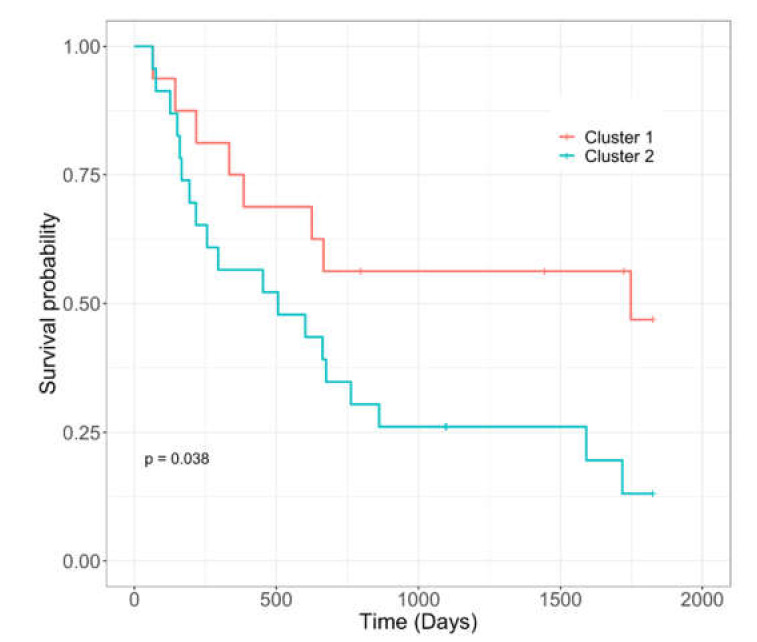
Survival analysis of HNSCC subtypes resulting from hierarchical clustering of HERV expression in tumor-adjacent normal tissue (Appendix A). Vertical lines indicate censored data. Curves were fit using the Kaplan–Meier model and statistical significance was calculated using the log-rank test.

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
