# Peer review of "Human Endogenous Retrovirus Expression Is Associated with Head and Neck Cancer and Differential Survival"

_viruses, 2020, doi:10.3390/v12090956_

Round 1

Reviewer 1 Report

In this bioinformatic study, the authors used data mining to characterize HERV expression in Head and Neck Squamous Cell Carcinoma (HNSCC). They used a Telescope, a recently developed program that is suitable for a locus-specific retrotranscriptomic analysis.

The study is descriptive. Nevertheless, it could provide useful source when the data are properly shared with the research community. The results should be presented in a way that the details of the analyses would be accessible to other researchers.

  1. Share the list of differentially expressed genes identified in the study (supplementary table X).
  2. Share locus specific information on data shown on Figure 2 (supplementary table Y).
  3. Share the locus specific information of the differentially expressed gene-HERV pair list, presented on the Circos plot (supplementary tableZ)
  4. Share the locus specific information on the clusters used for the survival study (supplementary table...)

Although, the authors claim that they provide the requested information in the revision. It is not uploaded. Thus, it is impossible the judge it. 

Under such circumstances, it is nor possible to accept the revised version.

Author Response

Reviewer 1

In this bioinformatic study, the authors used data mining to characterize HERV expression in Head and Neck Squamous Cell Carcinoma (HNSCC). They used a Telescope, a recently developed program that is suitable for a locus-specific retrotranscriptomic analysis.

The study is descriptive. Nevertheless, it could provide useful source when the data are properly shared with the research community. The results should be presented in a way that the details of the analyses would be accessible to other researchers.

Share the list of differentially expressed genes identified in the study (supplementary table X).

Share locus specific information on data shown on Figure 2 (supplementary table Y).

Share the locus specific information of the differentially expressed gene-HERV pair list, presented on the Circos plot (supplementary tableZ)

Share the locus specific information on the clusters used for the survival study (supplementary table...)

Although, the authors claim that they provide the requested information in the revision. It is not uploaded. Thus, it is impossible the judge it. 

Under such circumstances, it is nor possible to accept the revised version.

We are sorry to hear that you did not receive the revised supplementary materials. The revised Supplementary Table 1 contains the output of differential expression analysis of HERVs by DESeq2. Supplementary Table 2 contains the output of differential expression analysis of genes by DESeq2.  These tables contain all of the data used to create the circos plot as well as the data used to perform clustering for survival analysis. We have an additional 4 supplementary figures and an additional two tables.  All of these supplementary files were available for review as a single pdf from the journal web portal.  It is a large file with over 500 pages and over 9 mb of information.  It could be that the reviewer portal has issues distributing such a large file.  We are happy to set up a download option for the reviewer.  It is clear from the Reviewer 2 comments that this second reviewer was able to access the supplementary file.  Unfortunately, the journal submission process forces just a single pdf of all the supplemental material for reviewing which makes things a bit more complicated.

Reviewer 2 Report

The manuscript describes an analysis of the patterns of human endogenous retrovirus (HERV) expression in head and neck cancer tissue and normal-adjacent tissue.  The authors report abundant HERV expression in normal and cancer tissue along with substantial differential expression between the two.  HERV differential expression is characterized by overall up-regulation in cancer tissue and an enrichment for differentially expressed HERV-H sequences (distinct from previous reports that found differential expression of HERV-K in cancer tissue).  Both cancer and normal samples can be clearly resolved into two clusters by HERV expression profiles, but only the normal tissue HERV expression clusters differ with respect to cancer survival.  The results are taken to indicate the importance of HERV expression for studies of cancer.  The manuscript is technically solid and reports findings that are interesting in their own right; however, there is little by way of biological, functional or mechanistic explanation for the patterns observed here.  An effort to explore the broader implications of these findings, putting them in the context of previous studies even if somewhat speculative, would strengthen the manuscript.

Main points

1. The study relies on a program (pipeline) for the analysis of transposable element expression, Telescope, which was recently published by members of the same group. Telescope is distinguished by its ability to characterize the expression of repetitive HERV sequences at single locus resolution (an impressive feat). Based on what I can glean from the Telescope paper, the program can accurately assign multi-mapped RNA-seq tags to individual loci.  What is not clear is whether the HERV transcripts characterized via Telescope in this paper are bona fide HERV transcripts initiated from HERV promoters or whether they may be read-through, and potentially spurious/background, transcripts generated from nearby promoters in the genome.

2. The fact that normal tissue HERV expression clusters, and not cancer tissue clusters, are associated with differences in survival was surprising for me (or ‘counterintuitive’ as the authors put it in the discussion). The paper explains that this has in fact been found before for gene transcriptional profiles in a number of other cancers, thereby supporting their results. But I am still left with the question as to what this means?  The discussion could benefit from some consideration of the meaning and biological implications of this, however speculative, beyond simply stating that the same thing has been found previously for other cancers.

3. Similar to point #2, it is not clear from the manuscript what the implications of HERV upregulation in cancer tissue are, particularly in light of the survival difference association with HERV expression profiles in normal tissue. What does this mean? Does it simply reflect genome-wide dysregulation, or hypo-methylation, in cancer?  Have the authors checked whether there is an association with HERV cancer expression levels and survival, ie group cancer samples by overall HERV or family-specific (e.g. HERV-H) expression levels, or normal-cancer expression difference levels, and look for survival differences.  [Note that this would be distinct from the Euclidean distance based clustering method employed here to generate HERV expression groups].  In other words, is there any evidence that HERV up-regulation is associated with cancer survival differences?

4. The manuscript states that “Differentially expressed genes were found throughout the genome, similar to patterns observed for HERVs” and this is shown clearly in Figure 3. However, the level of resolution for Figure 3 doesn’t allow for a clear sense of whether HERVs and genes are co-expressed among the cancer and/or normal samples analyzed here.  Did the authors check this?  This may be a way to provide some putative functional insight into the observed patterns of HERV expression.

Minor points

5. Supplementary Figure 4 is reversed (the image is flipped).

6. The massive Supplementary Table 1 would be better distributed as an excel file (or CSV flatfile) to facilitate follow up analysis of these data.

Author Response

Reviewer 2

The manuscript describes an analysis of the patterns of human endogenous retrovirus (HERV) expression in head and neck cancer tissue and normal-adjacent tissue.  The authors report abundant HERV expression in normal and cancer tissue along with substantial differential expression between the two.  HERV differential expression is characterized by overall up-regulation in cancer tissue and an enrichment for differentially expressed HERV-H sequences (distinct from previous reports that found differential expression of HERV-K in cancer tissue).  Both cancer and normal samples can be clearly resolved into two clusters by HERV expression profiles, but only the normal tissue HERV expression clusters differ with respect to cancer survival.  The results are taken to indicate the importance of HERV expression for studies of cancer.  The manuscript is technically solid and reports findings that are interesting in their own right; however, there is little by way of biological, functional or mechanistic explanation for the patterns observed here.  An effort to explore the broader implications of these findings, putting them in the context of previous studies even if somewhat speculative, would strengthen the manuscript.

Main points

  1. The study relies on a program (pipeline) for the analysis of transposable element expression, Telescope, which was recently published by members of the same group. Telescope is distinguished by its ability to characterize the expression of repetitive HERV sequences at single locus resolution (an impressive feat). Based on what I can glean from the Telescope paper, the program can accurately assign multi-mapped RNA-seq tags to individual loci.  What is not clear is whether the HERV transcripts characterized via Telescope in this paper are bona fide HERV transcripts initiated from HERV promoters or whether they may be read-through, and potentially spurious/background, transcripts generated from nearby promoters in the genome.

There is always the possibility of some degree of background noise or read-through in RNA-seq data – affecting analysis of both genes and HERVs. However, we believe that our pipeline is robust and that we have taken appropriate steps to minimize spurious results. Firstly, as described in the simulation experiments in Bendall et al. (2019), Telescope is robust to sequencing error and minimizes detection of spurious sequences. After generating read counts with Telescope, we perform pre-filtering to remove background noise, keeping only HERVs with at least 5 reads in at least 2 samples. The normalization procedures performed by DESeq2 also further minimize the likelihood of identifying spurious differentially expressed loci.

While we have no way of knowing which promoters generated these transcripts, it is clear from previous work that HERVs are transcribed and in some cases, can perform a functional role. Functional characterization is necessary to ascertain whether the HERVs identified in our study perform an active role in cancer or whether they are simply bystanders.

  1. The fact that normal tissue HERV expression clusters, and not cancer tissue clusters, are associated with differences in survival was surprising for me (or ‘counterintuitive’ as the authors put it in the discussion). The paper explains that this has in fact been found before for gene transcriptional profiles in a number of other cancers, thereby supporting their results. But I am still left with the question as to what this means?  The discussion could benefit from some consideration of the meaning and biological implications of this, however speculative, beyond simply stating that the same thing has been found previously for other cancers.

We thank the reviewer for raising this point. Although it is not currently known why tumor-adjacent normal tissues can be predictive of cancer outcomes, several hypotheses exist. In particular, tumor-adjacent normal tissues have been shown to have unique transcriptional profiles compared to both true normal and tumor tissues. Due to their intermediate molecular state, tumor-adjacent normal tissues may be informative of immune or metabolic status. We have added text to the manuscript (lines 266-271) to discuss some of the current perspectives on tumor-adjacent normal tissue, and why this tissue type may provide more insight into patient survival than the tumor tissue itself. 

  1. Similar to point #2, it is not clear from the manuscript what the implications of HERV upregulation in cancer tissue are, particularly in light of the survival difference association with HERV expression profiles in normal tissue. What does this mean? Does it simply reflect genome-wide dysregulation, or hypo-methylation, in cancer? 

Understanding the implications of our findings will require follow-on experimental research. As we discuss in the introduction, HERVs may perform a variety of roles in cancer. Some HERVs may act in oncogenesis and have a negative effect on clinical outcomes; others may have a beneficial or protective role (see ref. 13 for review). Still others may play no role at all and may simply be bystanders. Therefore, we expect that HERVs identified as up-regulated in tumor tissue could play positive, negative, or neutral roles. We believe that this analysis provides insight into how patterns of HERV expression (including both up- and down-regulated HERVs) is associated with patient survival.  From our analysis, we have identified a variety of candidate HERVs that can be targeted for experimental studies.

Have the authors checked whether there is an association with HERV cancer expression levels and survival, ie group cancer samples by overall HERV or family-specific (e.g. HERV-H) expression levels, or normal-cancer expression difference levels, and look for survival differences.  [Note that this would be distinct from the Euclidean distance based clustering method employed here to generate HERV expression groups].  In other words, is there any evidence that HERV up-regulation is associated with cancer survival differences?

We believe that our approach best captures the differences in HERV expression levels. As shown in Figures 1 and 2, there are many HERVs that are downregulated in tumor tissue.  By performing clustering on all of the HERV expression data, we capture overall expression patterns which might include both upregulated and downregulated HERVs within a single family. Therefore, we don’t believe that aggregating by overall HERV or family-specific HERV expression levels would provide additional insight; rather, it would mask the more complex expression patterns observed at individual loci.

One way of looking at this question would be to assess whether any HERV family was enriched between the two survival groups. We did perform this analysis, but no significant enrichment was observed. We have added a sentence to this effect in the manuscript (lines 230-231).

  1. The manuscript states that “Differentially expressed genes were found throughout the genome, similar to patterns observed for HERVs” and this is shown clearly in Figure 3. However, the level of resolution for Figure 3 doesn’t allow for a clear sense of whether HERVs and genes are co-expressed among the cancer and/or normal samples analyzed here.  Did the authors check this?  This may be a way to provide some putative functional insight into the observed patterns of HERV expression.

We did not perform this analysis. Figure 3 is intended to demonstrate that differentially expressed HERVs are distributed throughout the genome. While we agree with the reviewer that coexpression analysis could be an interesting follow-up study, this is beyond the scope of our current work. In particular, special considerations would have to be made in order to perform this analysis due to the different methods used to quantify expression levels for genes versus HERVs.  We are actually developing methods to examine exactly this point at the moment.

Minor points

  1. Supplementary Figure 4 is reversed (the image is flipped).

The figure has been revised.

  1. The massive Supplementary Table 1 would be better distributed as an excel file (or CSV flatfile) to facilitate follow up analysis of these data.

We agree with the reviewer. It is our understanding that the files will be available in their original formats (csv) when the paper is published.

Round 2

Reviewer 1 Report

It is fine from my side, I accept the revision.

Reviewer 2 Report

I am satisfied with the authors' responses to my comments and recommend that the paper be accepted for publication.

This manuscript is a resubmission of an earlier submission. The following is a list of the peer review reports and author responses from that submission.

Round 1

Reviewer 1 Report

In this bioinformatic study, the authors used data mining to characterize HERV expression in Head and Neck Squamous Cell Carcinoma (HNSCC). They used a Telescope, a recently developed program that is suitable for a locus-specific retrotranscriptomic analysis.

The study is descriptive. Nevertheless, it could provide useful source when the data are properly shared with the research community. The results should be presented in a way that the details of the analyses would be accessible to other researchers.

  1. Share the list of differentially expressed genes identified in the study (supplementary table X).
  2. Share locus specific information on data shown on Figure 2 (supplementary table Y).
  3. Share the locus specific information of the differentially expressed gene-HERV pair list, presented on the Circos plot (supplementary tableZ)
  4. Share the locus specific information on the clusters used for the survival study (supplementary table...)

It is also not clear what the exact list of HERVs was used for the analysis.

General:

The figures are informative, but the text explaining them is not always clear.

Fig.2:

The color code for the different HERV families are too close to each other, difficult to distinguish. Use more contrasting colors.

Not clear what the connection between the Clusters shown on Figure 2 and the clusters use in the survival analysis used. Share locus information.

Fig3:

What are the HERV families that are most coregulated with differential gene expression?

Differentially expressed HERVs:

In addition to HERVH, HARLEQUIN was also specifically upregulated in the HNSCC samples, but is not presented on Fig. 2 and was not further commented. Neither commented on the significance of the HERVs, downregulated in cancer samples (HERVE, HML6, PRIMA4, and HERVEA).

Interpretation is weak:

It is clear that the mostly reported HERVK is not expressed differentially in HNSCC, but the explanation of DNA methylation is too general, and would not explain why certain elements are affected (e.g HERVH), while others are not. No methylation data shown.

From Fig. 2 there seem to be two different sets of HERVH loci. One that is specifically upregulated in cancer (even down in normal) and the other one that is specifically upregulated in the tumor-adjacent normal tissue. What is the difference between these elements or their loci? Any pattern? Please, share information.

Author Response

We thank the reviewers of our paper who have clearly read the paper thoroughly and have provided a number of insightful and helpful comments. We have strived to incorporate all of these comments and believe our manuscript is greatly improved as a consequence.  We thank the editor and reviewers for their significant time and effort in helping us more clearly and accurately communicate the exciting results of our research. Responses are written in italic.

Reviewer 1

In this bioinformatic study, the authors used data mining to characterize HERV expression in Head and Neck Squamous Cell Carcinoma (HNSCC). They used a Telescope, a recently developed program that is suitable for a locus-specific retrotranscriptomic analysis.

The study is descriptive. Nevertheless, it could provide useful source when the data are properly shared with the research community. The results should be presented in a way that the details of the analyses would be accessible to other researchers.

Share the list of differentially expressed genes identified in the study (supplementary table X).

We have added Supplementary Table 1, which contains the output of differential expression analysis of genes by DESeq2. 

Share locus specific information on data shown on Figure 2 (supplementary table Y).

We have added Supplementary Table 2, which contains the output of differential expression analysis of HERVs by DESeq2.

Share the locus specific information of the differentially expressed gene-HERV pair list, presented on the Circos plot (supplementary tableZ)

Full lists of differentially expressed genes and HERVs are now available in Supplementary Tables 1 and 2. 

Share the locus specific information on the clusters used for the survival study (supplementary table...)

The clustering was performed on all HERV expression results which are now available in Supplementary Tables 1 and 2.

It is also not clear what the exact list of HERVs was used for the analysis.

We have clarified language in the methods to indicate that the HERV annotation was obtained from previously published work.  See line 93.

General:

The figures are informative, but the text explaining them is not always clear.

We have revised the legend for Figure 2 according to reviewer comments.  See lines 217-220.

Fig.2:

The color code for the different HERV families are too close to each other, difficult to distinguish. Use more contrasting colors.

We have revised this figure with a most contrasting color palette for the HERV families.

Not clear what the connection between the Clusters shown on Figure 2 and the clusters use in the survival analysis used. Share locus information.

We have added clarifying information to the figure legend to indicate that the clustering was performed on these top 100 differentially expressed HERVs only.  Locus-specific information is available in Supplementary Table 2.  See lines 217-220.

Fig3:

What are the HERV families that are most coregulated with differential gene expression?

We did not perform coexpression analysis in this study.  This figure is intended to highlight the fact that differentially expressed HERVs are distributed throughout the genome. We have modified the legend to help make this more clear.

Differentially expressed HERVs:

In addition to HERVH, HARLEQUIN was also specifically upregulated in the HNSCC samples, but is not presented on Fig. 2 and was not further commented. Neither commented on the significance of the HERVs, downregulated in cancer samples (HERVE, HML6, PRIMA4, and HERVEA).

There is little known about these HERV families.  We have added a short paragraph highlighting the importance of future work on these HERV families in cancer.  See lines 288-291.

Interpretation is weak:

It is clear that the mostly reported HERVK is not expressed differentially in HNSCC, but the explanation of DNA methylation is too general, and would not explain why certain elements are affected (e.g HERVH), while others are not. No methylation data shown.

We did not perform methylation analysis in this study.  Our discussion of DNA methylation is limited to a previous study which identified decreased methylation on HERVH elements in tumor tissue in HNSCC.  To our knowledge, this is the only study that evaluated epigenetic changes on HERVs in HNSCC; therefore, we raise this as a potential link to the trends observed in our study and an area of potential future work.

From Fig. 2 there seem to be two different sets of HERVH loci. One that is specifically upregulated in cancer (even down in normal) and the other one that is specifically upregulated in the tumor-adjacent normal tissue. What is the difference between these elements or their loci? Any pattern? Please, share information.

We have added an ordered list of the HERV loci for Figure 2 in Supplementary Table 5. 

Reviewer 2 Report

Summary of the article:

Technical difficulties have long prevented locus-specific characterization of HERV expression. The authors use the tool Telescope on 43 paired tumor and adjacent normal tissue HNSC samples from TCGA to resolve this knowledge gap. They found 3000 expressed HERVs, of which 1078 HERVs are differentially expressed between the tissue types (normal vs tumor). These differentially expressed HERVs are enriched in HERVH family members. They clustered the samples based off total HERV expression and obtained two subclusters for each tissue type (two tumor clusters and two normal tissue clusters). When comparing survival probabilities across these subclusters, it was found that the normal tissue subclusters had differing survival probabilities. Furthermore, the authors noted that members of the HERVH family had an enriched expression in the cluster that had the improved survival outcome. This paper suggests that understanding HERV expression in different cancers and its influence in patient survival is a promising direction for future research.

Strengths:

Using the tool Telescope, the paper attempts to resolve a question that has been of interest but difficult to address due to technical difficulty: the question of identifying and characterizing ERV expression in cancer data. This paper represents an interesting attempt in the characterization of ERV elements in specifically HNSC (Head and Neck Cancer). The premise is exciting and the tool’s use to attempt to resolve this knowledge gap constitutes a contribution to the field.

Weaknesses:

While this paper is an interesting approach to a relevant problem, there remain some areas that would require improvement. One of the most important issues to resolve is the absence of several central figures and data analyses. Several key figures that are pivotal for the main claims of the paper, such as 1) the hierarchical clustering that identified the two normal sample clusters that differed in survival probability and 2) the data/figure that shows the enrichment of HERVH family members in the cluster with the higher survival probability, are missing from both the main article and the supplementary figures. Given that these represent the key findings of the paper, so much so that they are highlighted in the abstract as such, these figures should be included in the paper. Some of the findings are not quite novel and have been reported by others (e.g. HERVH expression in HNSCC in Kong et al. Nat Comms 2019 – a quite comprehensive analysis of retroviral elements in TCGA data, which should also be cited here). Although intriguing, the most promising finding of the paper – the ability of ERV expression in normal tissue to classify survival outcome, while expression information from tumors and protein coding gene expression does not – has only weak statistical support. This support should perhaps be subject to multiple testing correction, since the authors first try to associate survival with a different clustering approach and find no effect.

Specific Comments:

Comment 1 - Title: The title is slightly confusing and slightly misleading. The two main conclusions of the paper are 1) that HERV expression is higher in tumor than normal tissue, 2) HERV expression patterns in normal tissue may be indicative of survival. The first point is well documented in literature, so may not be desirable in the title. The second part of the title suggests that when HERVs are expressed, they determine survival – which I find misleading since the latter analysis is done in normal tissue, and it is not the general level of expression of ERVs but patterns of individual ERV expression that may be prognostic. In the authors later words (line 250) “Here, we show that HERV expression patterns as defined by hierarchical clustering in tumor adjacent normal tissue is related to altered probability of patient survival”. I understand that the title has to be terse, but it should more accurately reflect the main findings.  

Comment 2- Line 23-28: “The majority of differentially expressed HERVs were expressed at a higher level in tumor tissue. Differentially expressed HERVs were enriched in members of the HERVH family. Hierarchical clustering based on HERV expression in tumor adjacent normal tissue resulted in two distinct clusters with significantly different survival probability. Interestingly, members of the HERVH family were expressed at higher levels in the cluster with increased probability of survival rate.” It seems unexpected to include the HERVH expression as one of the main findings in the abstract. Throughout the paper, there is no discussion on HERVH expression in the normal tissue. If it is interesting, then there should be a mention of this finding somewhere in the body of the text. Also, a characterization of this result should be provided. Why was the HERVH expression interesting? Was it the only HERV family represented in normal tissue? Or the one with the highest fold change between clusters? Given that this is presented as one of the main points of the paper, then the analyses that brought about this observation should be presented, and the finding itself needs to be appropriately detailed and described. My impression is that  that the mention of HERVH is based primarily on prior literature, and not any findings that are presented here – in which case it should be removed from the abstract, since it is evident that when there are two clusters, expression of any element will be higher in one cluster than the other.

Comment 3 - line 28: “Together, these results indicate a significant role for HERVs in head and neck cancer”. Please remove this statement. These results by themselves do not indicate a role of HERVs in HNSC; they may well be indicative of tissue of origin or have a bystander (but possibly predictive) characteristic. Whether they may have some functional role, may be argued in the discussion, with appropriate justification from literature, but not stated in the abstract

Comment 4 - Line 39: “evidence of beneficial HERV effects“. This is a big claim, and it is not followed up with justification for this statement. How would they be beneficial? What are they doing that may be beneficial? I expect that the authors are referring to viral mimicry and immune response, but this is missing from the manuscript. I suggest this should be expanded here and more in depth in the discussion.

Comment 5 - Line 70-71: “Patients without a tobacco history data were categorized as smokers” Why would that be the default assumption? Would it not be more correct to exclude those patients without tobacco history data? How many patients out of the set of 43 were lacking in tobacco history data? That information is missing but is relevant for readers to know.

Comment 6 - Line 116: “HPV status was evaluated using PathoScope 2.0 [30].” Unless the HPV status info from TCGA was missing for some of the 43 patients part of this study, then why use this tool Pathoscope instead of relying on medical data from the TCGA? It seems like there is a lack of justification for this choice. The last author of this publication was also involved with the paper published for the development of PathoScope, which might explain the preference for using this tool, but if it is for some reason better than using the TCGA HPV annotation, then it should be mentioned.

Comment 7 - Lines 150-151: “Using Telescope [18], we identified 3520 HERVs expressed at a minimum of 5 counts in at least 2 samples across tumor and adjacent normal tissue from HNSCC cancer cases.” I agree that the readcount of at least 5 is a reasonable cutoff, but appropriate normalization needs to be considered to account for the variable number of total mapped reads across samples. If normalization was performed but not reported, please include and describe it.

Comment 8 - Line 175-177: “In both cases, one cluster was dominated by oral cavity tumors, which included mouth and tongue tumors, as well as tumors which overlapped the lip, oral cavity, and pharynx. The second cluster was split between oral cavity and laryngeal tumors” this is vague and although may information could be extracted from the figure, it would be easier to just cite the numbers in the text, e.g … “dominated by oral cavity tumors (x/y)… split between (w vs. z). “

Comment 9 - Line 182-183: “Although HPV status is an important risk factor in HNSCC, only two patients from this cohort were HPV positive.”  According to their analysis, 2 of the patients from their set of 43 are HPV positive. Given that it is known that HPV+ HNSCs are associated with better outcome, these patients should be removed from the analysis to not bias the results.

Comment 10 - Line 180-182: “Smoking status, gender, age at diagnosis, and tumor stage (early vs. late) were not significantly different between the two tumor clusters or between the two normal clusters.” This is not represented on the figures, but a more complete representation of the samples that includes this information would be helpful for readers.

Comment 11 - Line 183-185: “Although HPV status is an important risk factor in HNSCC, only two patients from this cohort were HPV positive. Interestingly, HPV positive patients in this analysis did not cluster together, indicating that other variables were driving the expression patterns in these cases.” The lack of clustering of the HPV+ patients is stated, but is not shown in any plots or diagrams in either the main article or in the supplementary figures. Please indicate in the heatmaps which samples are HPV(+).

Comment 12 - Figure 2: This may be a question of personal taste, but I’m not a fan of the color scheme usage in Figure 2: the tumor subsite, tumor site and tissue type all have the same color scheme (viridis). This is a bit inconvenient in terms of data visualization.

Comment 13 - Figure 2: In Figure 2, 3 of the normal samples cluster with the tumor samples in Figure 2, but this is not really discussed, described, explained or even commented on.

Comment 14 - Supplementary Figure 2: the distinction between the Tumor 1 vs Tumor 2 groups or the Normal 1 vs the Normal 2 groups in the PCA plot is not very clear. A potentially interesting follow-up figure that may distinguish these better would be if PCA plots were made with each tissue type separately?

Comment 15 - Line 213-216: “In order to evaluate the impact of HERV expression patterns on subtypes of HNSCC which are associated with patient survival probability, we performed hierarchical clustering on the HERV expression data followed by survival analysis. For this analysis, hierarchical clustering was performed separately on tumor and normal tissue, using all HERV expression data.” The figure showing this hierarchical clustering that permitted the identification of the two clusters with differential survival probability is not included in the paper or the supplementary figures. Given how important this clustering is to the main observations of this paper, it should definitely be provided, and the results of it (for example, the HERV expression present in each cluster) should be described before moving on to the Kaplan-Meier plot.

Comment 16 - Line 224-225: “Similarly, two clusters were identified from normal tissue. However, gender, tissue origin, and smoking status were not significantly different between these clusters (Chi-squared test, p < 0.05).” What about stage comparison? It is not mentioned here if it was significantly different between these clusters. Was it omitted by mistake or because it was significantly different? In either case, it should be mentioned.

Author Response

We thank the reviewers of our paper who have clearly read the paper thoroughly and have provided a number of insightful and helpful comments. We have strived to incorporate all of these comments and believe our manuscript is greatly improved as a consequence.  We thank the editor and reviewers for their significant time and effort in helping us more clearly and accurately communicate the exciting results of our research. Responses are written in italic.

Reviewer 2

Summary of the article:

Technical difficulties have long prevented locus-specific characterization of HERV expression. The authors use the tool Telescope on 43 paired tumor and adjacent normal tissue HNSC samples from TCGA to resolve this knowledge gap. They found 3000 expressed HERVs, of which 1078 HERVs are differentially expressed between the tissue types (normal vs tumor). These differentially expressed HERVs are enriched in HERVH family members. They clustered the samples based off total HERV expression and obtained two subclusters for each tissue type (two tumor clusters and two normal tissue clusters). When comparing survival probabilities across these subclusters, it was found that the normal tissue subclusters had differing survival probabilities. Furthermore, the authors noted that members of the HERVH family had an enriched expression in the cluster that had the improved survival outcome. This paper suggests that understanding HERV expression in different cancers and its influence in patient survival is a promising direction for future research.

Strengths:

Using the tool Telescope, the paper attempts to resolve a question that has been of interest but difficult to address due to technical difficulty: the question of identifying and characterizing ERV expression in cancer data. This paper represents an interesting attempt in the characterization of ERV elements in specifically HNSC (Head and Neck Cancer). The premise is exciting and the tool’s use to attempt to resolve this knowledge gap constitutes a contribution to the field.

Weaknesses:

While this paper is an interesting approach to a relevant problem, there remain some areas that would require improvement. One of the most important issues to resolve is the absence of several central figures and data analyses. Several key figures that are pivotal for the main claims of the paper, such as 1) the hierarchical clustering that identified the two normal sample clusters that differed in survival probability and 2) the data/figure that shows the enrichment of HERVH family members in the cluster with the higher survival probability, are missing from both the main article and the supplementary figures. Given that these represent the key findings of the paper, so much so that they are highlighted in the abstract as such, these figures should be included in the paper.

The hierarchical clustering used for survival analysis has been added to Supplementary Figures 3 and 4.  The second point was included in the abstract by error and has been removed (see line 30). 

Some of the findings are not quite novel and have been reported by others (e.g. HERVH expression in HNSCC in Kong et al. Nat Comms 2019 – a quite comprehensive analysis of retroviral elements in TCGA data, which should also be cited here).

We are happy to add any relevant papers to our discussion as appropriate.  The Kong et al. paper looked at all TEs and tend to lump them into ‘families’ in their results. They do point to a few HERVs, including HERVH and HERVK, as generally overexpressed in TCGA data. We now include this information and reference (see line 331).

Although intriguing, the most promising finding of the paper – the ability of ERV expression in normal tissue to classify survival outcome, while expression information from tumors and protein coding gene expression does not – has only weak statistical support. This support should perhaps be subject to multiple testing correction, since the authors first try to associate survival with a different clustering approach and find no effect.

We perform survival analysis on clustering of two different datasets (i.e., tumor expression and normal tissue expression).  Therefore, multiple testing correction is not necessary.

Specific Comments:

Comment 1 - Title: The title is slightly confusing and slightly misleading. The two main conclusions of the paper are 1) that HERV expression is higher in tumor than normal tissue, 2) HERV expression patterns in normal tissue may be indicative of survival. The first point is well documented in literature, so may not be desirable in the title. The second part of the title suggests that when HERVs are expressed, they determine survival – which I find misleading since the latter analysis is done in normal tissue, and it is not the general level of expression of ERVs but patterns of individual ERV expression that may be prognostic. In the authors later words (line 250) “Here, we show that HERV expression patterns as defined by hierarchical clustering in tumor adjacent normal tissue is related to altered probability of patient survival”. I understand that the title has to be terse, but it should more accurately reflect the main findings.  

We take the reviewer’s point here. Unfortunately, we have not been able to come up with something more clear and the reviewer has not made suggestions.  The title, as written, is indeed accurate and the abstract clearly states both main findings highlighted by the reviewer. So we have left the title as is.

Comment 2- Line 23-28: “The majority of differentially expressed HERVs were expressed at a higher level in tumor tissue. Differentially expressed HERVs were enriched in members of the HERVH family. Hierarchical clustering based on HERV expression in tumor adjacent normal tissue resulted in two distinct clusters with significantly different survival probability. Interestingly, members of the HERVH family were expressed at higher levels in the cluster with increased probability of survival rate.” It seems unexpected to include the HERVH expression as one of the main findings in the abstract. Throughout the paper, there is no discussion on HERVH expression in the normal tissue. If it is interesting, then there should be a mention of this finding somewhere in the body of the text. Also, a characterization of this result should be provided. Why was the HERVH expression interesting? Was it the only HERV family represented in normal tissue? Or the one with the highest fold change between clusters? Given that this is presented as one of the main points of the paper, then the analyses that brought about this observation should be presented, and the finding itself needs to be appropriately detailed and described. My impression is that  that the mention of HERVH is based primarily on prior literature, and not any findings that are presented here – in which case it should be removed from the abstract, since it is evident that when there are two clusters, expression of any element will be higher in one cluster than the other.

This point was included in the abstract by error and has been removed (see line 30).

Comment 3 - line 28: “Together, these results indicate a significant role for HERVs in head and neck cancer”. Please remove this statement. These results by themselves do not indicate a role of HERVs in HNSC; they may well be indicative of tissue of origin or have a bystander (but possibly predictive) characteristic. Whether they may have some functional role, may be argued in the discussion, with appropriate justification from literature, but not stated in the abstract

Point taken - we have removed this statement from the abstract (see line 30).

Comment 4 - Line 39: “evidence of beneficial HERV effects“. This is a big claim, and it is not followed up with justification for this statement. How would they be beneficial? What are they doing that may be beneficial? I expect that the authors are referring to viral mimicry and immune response, but this is missing from the manuscript. I suggest this should be expanded here and more in depth in the discussion.

This statement is summarizing existing literature on HERVs and their roles in cancer.  We have included a relevant citation that discusses the matter in more depth.  The point here is to emphasize the potential variable effect of HERV expression in cancers.  Our analysis does not enable us to draw any conclusions about whether HERVs are having harmful or beneficial effects on cancer; rather, we focus our analysis on the association of HERV expression patterns with variable patient outcomes.  Therefore, we do not believe additional discussion on specific mechanisms is warranted here.

Comment 5 - Line 70-71: “Patients without a tobacco history data were categorized as smokers” Why would that be the default assumption? Would it not be more correct to exclude those patients without tobacco history data? How many patients out of the set of 43 were lacking in tobacco history data? That information is missing but is relevant for readers to know.

We made this assumption to be consistent with methods used in the TCGA paper (https://doi.org/10.1038/nature14129).  We have added text to the methods indicating how many patients were missing tobacco history data, and to clarify that this method was chosen to be consistent with previous work.  See lines 76-77.

Comment 6 - Line 116: “HPV status was evaluated using PathoScope 2.0 [30].” Unless the HPV status info from TCGA was missing for some of the 43 patients part of this study, then why use this tool Pathoscope instead of relying on medical data from the TCGA? It seems like there is a lack of justification for this choice. The last author of this publication was also involved with the paper published for the development of PathoScope, which might explain the preference for using this tool, but if it is for some reason better than using the TCGA HPV annotation, then it should be mentioned.

HPV status was not available for all of the patients in this study.  In the TCGA paper, RNA-seq samples did not contain any clinical data regarding HPV status; instead, the authors mapped reads against a microbial database.  We were able to employ a similar approach using current viral databases and PathoScope, obtaining HPV status for all patients.  We now better justify this approach. See lines 126-127.

Comment 7 - Lines 150-151: “Using Telescope [18], we identified 3520 HERVs expressed at a minimum of 5 counts in at least 2 samples across tumor and adjacent normal tissue from HNSCC cancer cases.” I agree that the readcount of at least 5 is a reasonable cutoff, but appropriate normalization needs to be considered to account for the variable number of total mapped reads across samples. If normalization was performed but not reported, please include and describe it.

Normalization is performed as part of the standard DESeq2 pipeline.  We have added details to the methods for clarification.  See lines 99-102 and 118-122.

Comment 8 - Line 175-177: “In both cases, one cluster was dominated by oral cavity tumors, which included mouth and tongue tumors, as well as tumors which overlapped the lip, oral cavity, and pharynx. The second cluster was split between oral cavity and laryngeal tumors” this is vague and although may information could be extracted from the figure, it would be easier to just cite the numbers in the text, e.g … “dominated by oral cavity tumors (x/y)… split between (w vs. z). “

We have added these numbers to the text.  See lines 196-197 and 199.

Comment 9 - Line 182-183: “Although HPV status is an important risk factor in HNSCC, only two patients from this cohort were HPV positive.”  According to their analysis, 2 of the patients from their set of 43 are HPV positive. Given that it is known that HPV+ HNSCs are associated with better outcome, these patients should be removed from the analysis to not bias the results.

Given that only 2 of 43 patients were HPV positive, we do not expect that these patients would bias the results.  Furthermore, in both the clustering of top 100 differentially expressed HERVs (Figure 2) and the survival analysis (Figure 4), the HPV positive samples do not cluster together.  Therefore, we do not think that it is necessary to exclude these patients from the analysis. We have added text to the manuscript (see line 209) to indicate which samples were HPV positive to allow the reader to evaluate these differences.

Comment 10 - Line 180-182: “Smoking status, gender, age at diagnosis, and tumor stage (early vs. late) were not significantly different between the two tumor clusters or between the two normal clusters.” This is not represented on the figures, but a more complete representation of the samples that includes this information would be helpful for readers.

Given that Figure 2 is already quite complex, we have chosen not to represent these results on the figure.  Since these factors are not significant, we believe the current representation is sufficient. Access to the original metadata is available through TCGA, which we reference directly.

Comment 11 - Line 183-185: “Although HPV status is an important risk factor in HNSCC, only two patients from this cohort were HPV positive. Interestingly, HPV positive patients in this analysis did not cluster together, indicating that other variables were driving the expression patterns in these cases.” The lack of clustering of the HPV+ patients is stated, but is not shown in any plots or diagrams in either the main article or in the supplementary figures. Please indicate in the heatmaps which samples are HPV(+).

We have added text indicating which samples were HPV positive (see line 209). 

Comment 12 - Figure 2: This may be a question of personal taste, but I’m not a fan of the color scheme usage in Figure 2: the tumor subsite, tumor site and tissue type all have the same color scheme (viridis). This is a bit inconvenient in terms of data visualization.

We feel that adding additional color palettes would further complicate an already complex figure.  We prefer keeping the current color scheme.

Comment 13 - Figure 2: In Figure 2, 3 of the normal samples cluster with the tumor samples in Figure 2, but this is not really discussed, described, explained or even commented on.

We have added a couple of sentences addressing this observation.  See lines 204-208.

Comment 14 - Supplementary Figure 2: the distinction between the Tumor 1 vs Tumor 2 groups or the Normal 1 vs the Normal 2 groups in the PCA plot is not very clear. A potentially interesting follow-up figure that may distinguish these better would be if PCA plots were made with each tissue type separately?

We thank the reviewer for the suggestion.  We have evaluated the PCA plots separately, but they appear very similar to the aggregate plot.  We prefer to keep Supplementary Figure 2 with all samples for consistency with the PCA in the main figures.

Comment 15 - Line 213-216: “In order to evaluate the impact of HERV expression patterns on subtypes of HNSCC which are associated with patient survival probability, we performed hierarchical clustering on the HERV expression data followed by survival analysis. For this analysis, hierarchical clustering was performed separately on tumor and normal tissue, using all HERV expression data.” The figure showing this hierarchical clustering that permitted the identification of the two clusters with differential survival probability is not included in the paper or the supplementary figures. Given how important this clustering is to the main observations of this paper, it should definitely be provided, and the results of it (for example, the HERV expression present in each cluster) should be described before moving on to the Kaplan-Meier plot.

We have added Supplementary Figure 3A and Supplementary Figure 4 showing the clustering for both tumor and normal tissue samples.

Comment 16 - Line 224-225: “Similarly, two clusters were identified from normal tissue. However, gender, tissue origin, and smoking status were not significantly different between these clusters (Chi-squared test, p < 0.05).” What about stage comparison? It is not mentioned here if it was significantly different between these clusters. Was it omitted by mistake or because it was significantly different? In either case, it should be mentioned.

We have added the result by stage (not significant).  See lines 250 and 257.

Round 2

Reviewer 2 Report

My previous two main concerns were the

  • perceived lack of novelty of the first main claim of the manuscript: “HERV Expression is associated with head and neck cancer” and
  • weak support for the second claim that “HERV expression is associated with differential survival”

Regarding the first point, I asked for clarification and discussion of a recent paper by Kong et al., a pan-cancer analysis, which among other findings reported increased expression of HERVH elements in HNSCC tumors compared to matched normal tissue. I felt that the response of the authors was inadequate. First, the authors agreed to cite the article, but it is cited in a way that does not fully credit the findings I pointed out. “This loss of methylation could contribute to the increased expression of HERVH elements that we observe in this study in tumor tissue relative to adjacent-normal tissue [50].” It almost seems like the current manuscript (Kolbe et al.) made this important observation regarding HERVH expression, and Kong et al findings can shed some light on the mechanisms. This is not accurate, as explained more below. The second part of the authors’ reply seemed to somewhat discount the Kong et al article as “lumping elements into families” and “pointing to a few HERVs as generally overexpressed in TCGA data…”.

I decided that I have to read the Kong et al article more carefully. Both of the statements the authors make are incorrect. There is no “lumping” taking place. Kong et al use a pretty smart approach that uses EM to assign ambiguous reads to unique loci based on uniquely mapped reads. Thus, expression of every TE locus is estimated. It is true, that following this initial mapping, Kong et al also measure overall expression per TE family, but this is only a matter of order of operations. Kolbe et al go from locus expression -> differential expression analysis - > aggregating DE loci into families, Kong et al. do locus expression -> aggregating TEs into families - > differential expression analysis. The result is the same in both cases: HERVH expression is significantly higher in HNSCC tumors than adjacent tissue. And it is not true – as the authors claim in the reply – that Kong et al do this in general TCGA dataset; they show this explicitly for HNSCC (Figure 2e, and in detail in supplementary figure 2 g,h). The Kong et al paper is also interesting in many other ways, since they subivide the TEs according to genic and intergenic locations, show general relationships of TE expression and local DNA methylation, and immune response of the tumor. But the main point is that along the way, they already show that “HERV expression is associated with head and neck cancer”.

My second point was not addressed appropriately. The authors are incorrect in stating that no multiple testing correction is required. The simplest way to look at it is that they are testing association of survival with 1) tumor-based clustering, 2) adjacent-normal based clustering. That’s at least two tests that are explicitly presented in the paper. Unless the authors are prepared to state that they are only interested in the normal analysis and would have not reported association of survival to tumor clusters (if found significant), what they are performing is the definition of multiple testing.

Given the authors reply and further reading of existing literature, I don’t feel that the findings presented in this manuscript are either novel enough or sufficiently strongly supported to recommend publication.